# Assessment of the consistency of health and demographic surveillance and household survey data: A demonstration at two HDSS sites in The Gambia

**Momodou Jasseh**[1]*, **Anne J. Rerimoi**[2], **Georges Reniers**[3], **Ian M. Timæus**[3,4]

**1** Medical Research Council Unit The Gambia at London School of Hygiene & Tropical Medicine, Fajara, The Gambia, **2** Department of Women and Children's Health, King's College, London, United Kingdom, **3** Department of Epidemiology and Population Health, Population Studies Group, London School of Hygiene & Tropical Medicine, London, United Kingdom, **4** Centre for Actuarial Research, University of Cape Town, Cape Town, South Africa

* Momodou.Jasseh@lshtm.ac.uk

## Abstract

### Objective

To assess whether an adapted Demographic and Health Survey (DHS) like cross-sectional household survey with full pregnancy histories can demonstrate the validity of health and demographic surveillance (HDSS) data by producing similar population structural characteristics and childhood mortality indicators at two HDSS sites in The Gambia–Farafenni and Basse.

### Methods

A DHS-type survey was conducted of 2,580 households in the Farafenni HDSS, and 2,907 in the Basse HDSS. Household members were listed and pregnancy histories obtained for all women aged 15–49. HDSS datasets were extracted for the same households including residency episodes for all current and former members and compared with the survey data. Neonatal (0–28 days), infant (<1 year), child (1–4 years) and under-5 (< 5 years) mortality rates were derived from each source by site and five-year periods from 2001–2015 and by calendar year between 2011 and 2015 using Kaplan–Meier failure probabilities. Survey-HDSS rate ratios were determined using the Mantel-Haenszel method.

### Results

The selected households in Farafenni comprised a total population of 27,646 in the HDSS, compared to 26,109 captured in the household survey, implying higher coverage of 94.4% (95% CI: 94.1–94.7; *p*<0.0001) against a hypothesised proportion of 90% in the HDSS. All population subgroups were equally covered by the HDSS except for the Wollof ethnic group. In Basse, the total HDSS population was 49,287, compared to 43,538 enumerated in the survey, representing an undercount of the HDSS by the survey with a coverage of

**Data Availability Statement:** Data from the Farafenni Health and Demographic Surveillance System (FHDSS) are available on ishare2-

INDEPTH repository - "Farafenni INDEPTH Core Dataset 1982-2015 (Release 2018), which is provided by the INDEPTH Network Data Repository. Users may access this data by following the URL: www.indepth-network.org, DOI:10.7796/INDEPTH.GM011.CMD2015.v2. Data from the Basse Health and Demographic Surveillance System (BHDSS) and the DHS-type household survey data are available from the Medical Research Council Unit The Gambia at London School of Hygiene and Tropical Medicine (MRCG@LSHTM) through the Scientific Coordinating Committee (scc@mrc.gm) and subject to approval by the Joint MRCG/Gambia Government Ethics Committee for researchers who meet the criteria for access to confidential data.

**Funding:** The Farafenni and Basse HDSS sites are supported by Medical Research Council Unit The Gambia at London School of Hygiene and Tropical Medicine (MRCG@LSHTM). The cross-sectional sample household survey was supported by Nagasaki University (NU), Japan, through an agreement, MRC Internal Reference number 506518. Both MRCG@LSHTM and NU had no role in the study design, data collection and analysis, decision to publish, or preparation of the manuscript.

**Competing interests:** The authors have declared that no competing interests exist.

88.3% (95% CI: 88.0–88.6; $p = 1$). All sub-population groups were also under-represented by the survey. Except for the neonatal mortality rate for Farafenni, the childhood mortality indicators derived from pregnancy histories and HDSS data compare reasonably well by 5-year periods from 2001–2015. Annual estimates from the two data sources for the most recent quinquennium, 2011–2015, were similar in both sites, except for an excessively high neonatal mortality rate for Farafenni in 2015.

## Conclusion

Overall, the adapted DHS-type survey has reasonably represented the Farafenni HDSS database using population size and structure; and both databases using childhood mortality indicators. If the hypothetical proportion is lowered to 85%, the survey would adequately validate both HDSS databases in all considered aspects. The adapted DHS-type sample household survey therefore has potential for validation of HDSS data.

## Introduction

Despite the significant methodological improvements made in generating demographic information in most low and middle-income countries (LMICs), gaps remain in terms of quality and periodicity of existing data collection methods. Thus, they remain inadequate to support world class continuous scientific and socio-economic investigations to inform policy and influence practices for improving quality of life. In the absence of functional national vital registration systems, public health researchers in the global south have over the past two decades become increasingly dependent on health and demographic surveillance systems (HDSSs) for measurement of the impact of controlled interventions in communities and deriving reliable demographic indicators of interest, which can only otherwise be obtained from periodic cross-sectional Demographic and Health Surveys (DHS) and Multiple Indicator Cluster Surveys (MICS), or censuses. As a result, HDSS sites in sub-Saharan Africa have been instrumental in contributing evidence to demonstrate progress and achievements against the millennium development goals (MDGs) for instance, especially MDG4 [1, 2].

Despite criticisms levelled against them regarding their unrepresentativeness of national populations [3] and counter arguments in their favour [4, 5], dependence on these platforms for health and socio-economic research will continue for the foreseeable future. They will be key for measurement of targets associated with Sustainable Development Goal 3 (ensuring healthy lives). They also constitute the only appropriate resource to support medical research and large field trials in most LMICs that require precise numbers of cases and corresponding denominator counts for accurate measurement of incidence or prevalence rates of interest. But, like any other source of demographic information in LMICs, an HDSS is prone to both random and systematic errors. Most of these are usually identified and corrected using locally-designed check algorithms and data plausibility assessments. Moreover, other methodological simulations have demonstrated that HDSSs are sufficiently robust to withstand introduced random errors in excess of 20% [6]. Notwithstanding such internal integrity claims, HDSS databases need to be systematically and independently assessed and evaluated on a regular basis if they are to be relied on to generate evidence to influence policy in the southern world.

There are no set methods for independently assessing HDSS data for external and internal structural consistency. The traditional approach to HDSS data evaluation has been through an

independent re-census of the entire demographic surveillance area, usually culminating in a dataset that can only be used to confirm household membership and population sizes of settlements and the entire surveillance area. The occurrence of key demographic events such as births and deaths, which are crucial to HDSS operations and measurement of demographic indicators and trends, cannot be ascertained by such a simple recount of residents. Other previous attempts to assess the quality of HDSS data and derived demographic indicators have focussed mainly on comparisons with data from Demographic and Health Survey (DHS) conducted in regions where the HDSS sites are located [7, 8]. As part of the recent "Every Newborn–INDEPTH" study (EN-INDEPTH) [9], HDSS data from five sites in Africa and Asia were used to assess data quality indicators for births and deaths data collected through a survey [10].

This objective of this study is to establish whether a sample DHS-type cross-sectional household survey conducted in a demographic surveillance area, collecting full pregnancy histories from women aged 15–49, can be used to assess routine HDSS data in terms of population counts and characteristics, and childhood mortality indicators. It compares household survey data with HDSS data from two demographic surveillance sites in The Gambia–the Farafenni and Basse Health and Demographic Surveillance Systems (FHDSS and BHDSS).

## Methods

### Study areas and populations

The Farafenni and Basse HDSS sites are located in middle and eastern Gambia respectively as shown in Fig 1. The Farafenni HDSS is described in detail elsewhere [11]. It is situated in the North Bank Region of The Gambia about 170 km from the capital city, Banjul; covering an area spanning 32km east and 22 km west of Farafenni Town. It was set up in October 1981 and initially comprised two clusters of 42 rural villages and hamlets located at least 10 km east

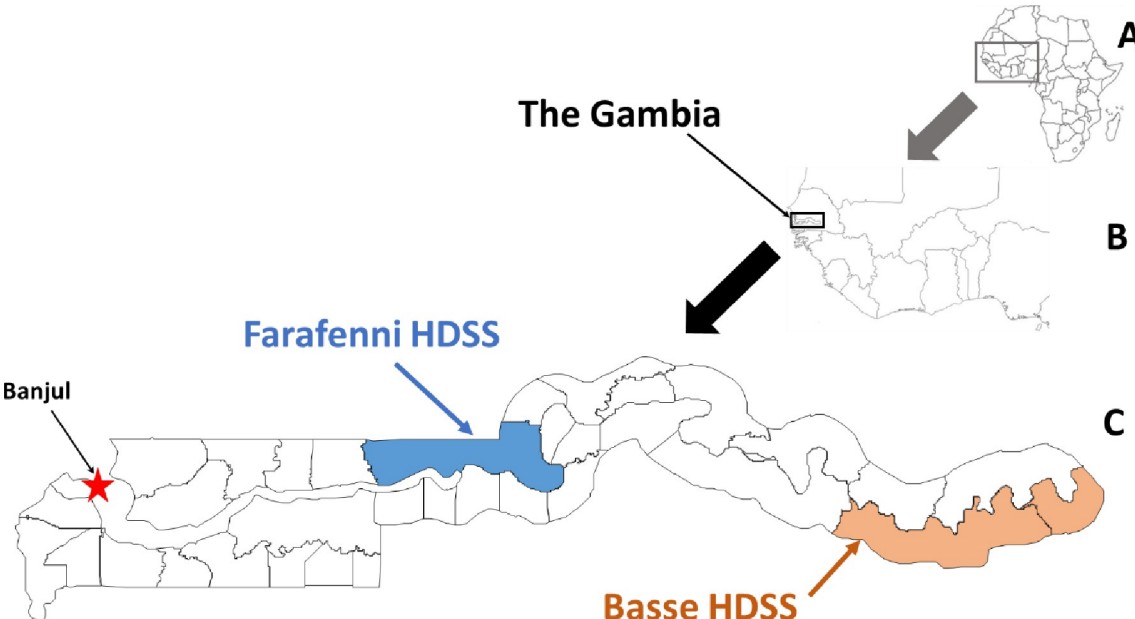

**Fig 1. The geographical locations of the Farafenni and Basse HDSS sites in The Gambia.** A and B republished from [https://d-maps.com/m/africa/afrique/afrique10.gif] [13] and [https://d-maps.com/m/africa/west/west06.gif] [14] under a CC BY license, with permission from d-maps.com; C from https://commons.wikimedia.org/wiki/File:Gambia_districts.png [15].

and west of Farafenni town and a total population of about 12,000. The HDSS was expanded in 2002 to include Farafenni town and 23 settlements within a 5km radius of the town, referred to as the urban surveillance area. It prospectively follows up a total population of about 58,000 who are predominantly subsistence farmers with an annual average income per capita of less than UD$150 [11]. The site has supported cutting-edge medical, public health and demographic research since its inception; and three decades of surveillance in the area has documented significant declines in malaria transmission [12] and child mortality [1]. Surveillance was interrupted for a period of 13 months between 1st March 2008 and 31st March 2009.

The Basse HDSS covers all 224 villages and hamlets on the south bank of the Upper River Region, the eastern-most part of The Gambia. It covers an area of 1,111 sq Km and is located between 350 and 450 Km from the capital city. After an initial census conducted between September and November 2006 with an enumerated population of 136,387, routine demographic surveillance commenced in July 2007 [16, 17]. The population is presently about 190,000. Like in Farafenni, most residents are subsistence farmers with similar level of per capita income.

Both sites are characterised by high fertility levels with total fertility rates of more than 5 children per woman; and have large illiterate populations with more than 50% of females and over 40% of males aged 6 years and over in the administrative regions they are located in having never been to school [18]. Rural-urban migration is also a common feature in these rural regions. The 2013 National Population Census documented a net migration rate of -28% for the North Bank Region, and -13.4% for Upper River Region [19]. The main urban municipalities of Banjul and Kanifing serve as the main destinations for young males seeking employment opportunities, and females joining their husbands.

## Health and demographic surveillance procedures

Surveillance procedures in Farafenni have been described in detail elsewhere [11]. The same procedures apply in Basse. Briefly, each system places four primary units under surveillance in their respective geographically demarcated areas and updates them accordingly in four-monthly intervals. These units are the settlements, compounds, households and individuals. A settlement comprises one or more compounds, which are usually demarcated by a fence; and a compound consists of one or more households. In both surveillance sites, a household is defined as a person or group of persons living in the same house or compound and sharing the same cooking arrangements. All settlements, compounds, households and individuals have unique identification numbers. New settlements, though rare, and compounds are enumerated as and when they appear; whilst the status of households and residency of individual members are updated during each four-monthly round of visits conducted by trained fieldworkers, who are predominantly male secondary school graduates, and aged between 20 and 45 years. The household head is the primary respondent during these updates, or an appropriate representative in his or her absence. Other household members are called on to provide other person-specific information such as pregnancy status and children's vaccination records. In addition, resident village reporters recruited and trained on the recording of demographic events within their communities, keep records of births, deaths and migrations in and out of their settlements. This information is used by the fieldworkers to cross-check data collected during compound and household visits. The information was collected using household registration books up to 30th September 2015 in Farafenni and 29th February 2016 in Basse. Electronic data capture was then rolled out at both sites using computer tablets and customised software. Fieldworkers also record marriages and pregnancies. The latter are followed up to termination to confirm pregnancy outcomes and ascertain early neonatal and infant deaths.

## Sample household survey

A DHS-type sample household survey was conducted between 10th December 2015 and 30th April 2016 by 10 enumerators in the Farafenni demographic surveillance area (DSA); and between 17th December 2015 and 4th June 2016 by 21 enumerators in the Basse DSA. Because of specific requirements of another study on neonatal mortality on which the surveys were based, half of all active households in the FHDSS and a quarter of households in the BHDSS as at 30th November 2015, evenly distributed across the respective surveillance areas, were randomly selected to participate. These represented a target of 2,862 households in Farafenni, and 2,780 in Basse. Only female enumerators, who were secondary school graduates in their early twenties, were recruited for cultural reasons to facilitate sufficient probing to distinguish between stillbirths and early neonatal deaths, as well as between abortions, miscarriages and stillbirths. None of the interviewers previously worked in the HDSS. The unique identification numbers of selected households were extracted from the HDSS databases and served as the only link between the survey and the HDSS to ensure independence between the two data sources. The same definition of household used in the HDSS was adopted by the enumerators for the sample survey.

The survey involved the administration of versions of the DHS household form, responded to by the household head or appropriate representative, and individual woman's questionnaire adapted for use in this context [20]. In the household form, all resident members were listed and their relevant socio-demographic details recorded, including parental survival and residence. Individuals who had not spent two of the last three months in the household were excluded from further analysis. All women aged between 15 and 49 years listed in the household schedule were identified and a separate woman's questionnaire administered to each woman personally. This included the standard DHS questions on reproduction, and detailed pregnancy and sibling histories. In obtaining the full pregnancy history of a woman, the enumerator asked about all pregnancies the woman had ever had, starting with the most recent and working back to the first pregnancy. For each pregnancy, the outcome (i.e. live birth, stillbirth, miscarriage or abortion) was recorded, together with the month and year of termination. In the case of live births, survival status of the child was established, with age at last birthday if still alive, or age at death if the child had died. All reported pregnancies that resulted in live births, and terminated between 1st January 2001 and 31st December 2015, are included in the analysis.

## Data and statistical analysis

Since the assessment requires comparison of data from the same households, it was necessary to ensure that all households included in the survey could be linked to the correct household in the HDSS database using the unique household identification number, HHID. The HHID consists of eight digits in Farafenni, and eleven in Basse. A total of 2,652 households were surveyed in Farafenni and 2,993 in Basse and subjected to a systematic search to ensure that they correspond to the same households in the respective HDSS databases. Eventually, 72 households (2.7%) were dropped in Farafenni, 52 of which were with HHIDs that could not be reliably identified in the FHDSS database due to transcription error, and the rest were duplicates. The corresponding situation in Basse was 86 dropped households (2.9%), 74 not meeting the requirements for inclusion. Consequently, 2,580 households in Farafenni and 2,907 households in Basse were confirmed for inclusion in the analysis. The pregnancy histories of all women aged 15–49 in these households were collated and only those that terminated in live births were selected to generate a child-based dataset for each site for mortality estimation. No attempt was made to link mothers to their children between the two data sources.

The final lists of household identification numbers were also used to extract the relevant episodes and events associated with all household members, present and past, from the HDSS

databases, censoring all episodes at 30th April 2016 in Farafenni and 4th June 2016 in Basse. These were regarded as the comparable HDSS datasets to be evaluated by the survey data in terms of population size, structure and characteristics, and childhood mortality levels and trends.

The survey and HDSS data were compared at two levels to ascertain the external and internal consistencies of the two data sources respectively. For the first level comparison, the coverage of the HDSS by the survey, both for total population counts and proportions of selected characteristics, was assessed to determine whether it adequately captured the HDSS population at the time of the fieldwork. The one-sample proportion test was used [21], applying the survey variable counts as sample size and the respective proportions they represent to the corresponding HDSS counts. A hypothesised proportion was required to perform the test, i.e. a threshold level considered to be the minimum acceptable coverage of the HDSS by the survey. None is recommended in the literature of HDSS and DHS methodologies. However, an *a priori* level of 90% was proposed and used in the analysis for two reasons. The first is the difference in operational definitions used in the two methods of data collection, especially with respect to residency status. For instance, household members who have been away for periods of up to four months and still considered residents by HDSS definition are likely to be missed in the count of residents in the household survey. Secondly, misapplication of the adopted definition of a household was more likely to divide unusually large households into smaller nuclear family units in the survey. Both situations result in an undercount of household members in the survey; and therefore, an overall coverage of at least 90% can be considered an acceptable capture of the HDSS in general. Therefore, with a survey coverage, $p$, of the HDSS and a hypothesised proportion of 0.9, a null hypothesis, $H_0$: $p \leq 0.9$, was tested against an alternative hypothesis, $H_A$: $p > 0.9$ at 95% confidence level using the $z$-test statistic.

In the second or internal consistency comparison, the differences between the proportions of the selected population characteristics from both sources were compared using the two-sample proportion test [21]. An assumed null hypothesis of $H_0$: $p_s = p_h$ was tested against an alternative hypothesis of $H_A$: $p_s \neq p_h$ at 95% confidence level using the $z$-test statistic, where $p_s$ is the survey proportion and $p_h$ the HDSS proportion. For the purpose of this analysis, broadly defined age groups by sex (i.e., 0, 1–4, 5–14, 15–29, 30–49, 50–64, and 65+) and ethnic group were selected as the population characteristics to focus on for the main reason that they are likely to be correctly reported in both the survey and HDSS.

The comparison of childhood mortality indicators derived from the two sources of data for both sites was based on calendar year estimates covering the 15-year period between 2001 and 2015. Childhood mortality indicators and rate ratios were estimated using survival analysis in Stata 17 [22]. Mortality rates from each data source were calculated as the number of deaths per thousand live births and were determined for each period and year using Kaplan–Meier failure probabilities for each of the age groups of interest that constitute measures of childhood mortality. These are defined as: (i) deaths within the first month of life (i.e. <1 month) for neonatal mortality; (ii) deaths in the first year of life (i.e. <1 year) for infant mortality; (iii) deaths between exact ages one and four years (i.e. 1–4 years) for child mortality; and (iv) deaths between birth and exact age five years (<5 years) for under-5 mortality. For each age group, period and recent year survey-HDSS rate ratios were determined using the Mantel-Haenszel method and $p$-values reported at 95% confidence level.

## Ethics statement

The Farafenni and Basse HDSSs have approval from the Joint MRCG/Gambia Government Ethics Committee to conduct continued surveillance. The same committee approved the

instruments used in the household surveys at both sites, and for which consent was sought and documented from household heads and eligible women prior to the administration of the respective questionnaires.

# Results

## Comparison of population indicators

The household size characteristics of the sampled households in both sites are presented in Table 1. In each of the sites, the two data sources show similar size characteristics, with households tending to be much larger in Basse than Farafenni. Compared with the HDSS, the survey undercounted the sizes of 13% and 18% of households in Farafenni and Basse, respectively. Collectively, the Farafenni households yielded a total population of 27,646 in the HDSS, compared to 26,109 captured in the household survey, implying a coverage of 0.944 (95% CI: 0.941–0.947) (see Table 2). Applying the one-sample proportion test against the hypothesised proportion of 0.9, there is no evidence to suggest that the survey in Farafenni inadequately

**Table 1. Descriptive statistics on households and comparison of reported household sizes by site and source.**

| FARAFENNI | | | | | | |
|---|---|---|---|---|---|---|
| | | | **Survey** | | **HDSS** | |
| No. of households | | | 2,580 | | 2,580 | |
| Household size range | | | 1–58 | | 1–61 | |
| Mean household size | | | 10 | | 11 | |
| Median household Size | | | 5 | | 5 | |
| *Comparison of reported household size ranges*: | | | | | | |
| | | | HDSS Household Size | | | |
| Survey Household Size | 1–10 | 11–20 | 21–40 | 41–60 | 60 + | Survey- undercounted HHs* |
| 1–10 | **1,357** | 235 | 12 | - | - | 247 |
| 11–20 | 124 | **610** | 79 | 2 | - | 81 |
| 21–40 | 4 | 35 | **114** | 5 | - | 5 |
| 41–60 | - | - | 1 | **1** | 1 | 1 |
| 60 + | - | - | - | - | **0** | Total: 334 *(13%)* |
| Survey-overcounted HHs | 128 | 35 | 1 | 0 | | Total: 164 *(6%)* |
| BASSE | | | | | | |
| | | | **Survey** | | **HDSS** | |
| No. of households | | | 2,907 | | 2,907 | |
| Household size range | | | 1–117 | | 1–130 | |
| Mean household size | | | 15 | | 17 | |
| Median household Size | | | 9 | | 11 | |
| *Comparison of reported household size ranges*: | | | | | | |
| | | | HDSS Household Size | | | |
| Survey Household Size | 1–10 | 11–20 | 21–40 | 41–60 | 60 + | Survey- undercounted HHs |
| 1–10 | **1,147** | 212 | 32 | 8 | 2 | 254 |
| 11–20 | 121 | **532** | 153 | 21 | 8 | 182 |
| 21–40 | 11 | 73 | **379** | 47 | 20 | 67 |
| 41–60 | 1 | 1 | 24 | **63** | 14 | 14 |
| 60 + | 1 | - | 1 | 5 | **31** | Total: 517 *(18%)* |
| Survey-overcounted HHs | 134 | 74 | 25 | 5 | | Total: 238 *(8%)* |

* *HHs = Households.*

**Table 2. Survey coverage of the HDSS by site and population characteristics; and outputs of one- and two-sample proportion tests.**

| Site and population characteristic | Survey | (%) | HDSS | (%) | Survey coverage of HDSS | (95% CI*) | z-score | p-value ($H_0$ vs $H_a$: $p > p_0$) | Differ-ence | (95% CI) | z-score | p-value |
|---|---|---|---|---|---|---|---|---|---|---|---|---|
| **FARAFENNI** | | | | | | | | | | | | |
| Total Population | 26,109 | | 27,646 | | 0.944 | (0.941, 0.947) | 24.39 | < **0.001** | | | | |
| Age and Sex | | | | | | | | | | | | |
| Male | 11,780 | (45.1) | 12,572 | (45.5) | 0.937 | (0.933, 0.941) | 13.83 | < **0.001** | -0.004 | (-0.017, 0.009) | -0.630 | **0.531** |
| 0 | 1,024 | (8.7) | 430 | (3.4) | 2.381 | - | - | - | 0.053 | (0.028, 0.077) | 3.570 | < **0.001** |
| 1–4 | 1,996 | (16.9) | 2,069 | (16.5) | 0.965 | (0.957, 0.973) | 9.86 | < **0.001** | 0.004 | (-0.019, 0.027) | 0.380 | **0.707** |
| 5–14 | 3,488 | (29.6) | 3,843 | (30.6) | 0.908 | (0.899, 0.917) | 1.65 | **0.049** | -0.010 | (-0.031, 0.011) | -0.890 | **0.371** |
| 15–29 | 2,422 | (20.6) | 2,967 | (23.6) | 0.816 | (0.802, 0.830) | -15.25 | **1.000** | -0.030 | (-0.053, -0.008) | -2.670 | **0.008** |
| 30–49 | 1,586 | (13.5) | 1,912 | (15.2) | 0.829 | (0.812, 0.846) | -10.35 | **1.000** | -0.018 | (-0.041, 0.006) | -1.470 | **0.142** |
| 50–64 | 822 | (7.0) | 894 | (7.1) | 0.919 | (0.901, 0.937) | 1.89 | **0.029** | -0.001 | (-0.026, 0.023) | -0.110 | **0.914** |
| 65+ | 442 | (3.8) | 457 | (3.6) | 0.967 | (0.951, 0.983) | 4.77 | < **0.001** | 0.001 | (-0.023, 0.026) | 0.090 | **0.926** |
| Female | 14,229 | (54.5) | 15,074 | (54.5) | 0.944 | (0.940, 0.948) | 18.01 | < **0.001** | 0.000 | (-0.011, 0.011) | 0.000 | **1.000** |
| 0 | 875 | (6.1) | 356 | (2.4) | 2.458 | - | - | - | 0.038 | (0.015, 0.060) | 2.750 | **0.006** |
| 1–4 | 1,816 | (12.8) | 1,917 | (12.7) | 0.947 | (0.937, 0.957) | 6.86 | < **0.001** | 0.001 | (-0.021, 0.022) | 0.070 | **0.942** |
| 5–14 | 3,810 | (26.8) | 4,097 | (27.2) | 0.930 | (0.922, 0.938) | 6.40 | < **0.001** | -0.004 | (-0.024, 0.016) | -0.400 | **0.689** |
| 15–29 | 3,926 | (27.6) | 4,254 | (28.2) | 0.923 | (0.915, 0.931) | 5.00 | < **0.001** | -0.006 | (-0.026, 0.013) | -0.630 | **0.526** |
| 30–49 | 2,313 | (16.3) | 2,914 | (19.3) | 0.794 | (0.779, 0.809) | -19.07 | **1.000** | -0.031 | (-0.051, -0.010) | -2.870 | **0.004** |
| 50–64 | 1,062 | (7.5) | 1,045 | (6.9) | 1.016 | - | - | - | 0.005 | (-0.017, 0.027) | 0.470 | **0.637** |
| 65+ | 427 | (3.0) | 491 | (3.3) | 0.870 | (0.840, 0.900) | -2.22 | **0.987** | -0.003 | (-0.025, 0.020) | -0.220 | **0.824** |
| Not Stated | 100 | | - | | | | | | | | | |
| Ethnic Group | | | | | | | | | | | | |
| Wollof | 9,695 | (37.1) | 10,821 | (39.1) | 0.896 | (0.890, 0.902) | -1.39 | **0.917** | -0.020 | (-0.033, -0.007) | -2.940 | **0.003** |
| Mandinka | 8,989 | (34.4) | 9,406 | (34.0) | 0.956 | (0.952, 0.960) | 18.10 | < **0.001** | 0.004 | (-0.010, 0.018) | 0.570 | **0.568** |
| Fula | 5,124 | (19.6) | 5,621 | (20.3) | 0.912 | (0.905, 0.919) | 3.00 | **0.001** | -0.007 | (-0.022, 0.008) | -0.910 | **0.365** |
| Other | 2,000 | (7.7) | 1,798 | (6.5) | 1.112 | - | - | - | 0.012 | (-0.004, 0.028) | 1.430 | **0.151** |
| Not stated | 301 | (1.2) | | | | | | | | | | |
| **BASSE** | | | | | | | | | | | | |
| Total Population | 43,538 | | 49,287 | | 0.883 | (0.880, 0.886) | -12.58 | **1.000** | | | | |
| Age and Sex | | | | | | | | | | | | |
| Male | 19,830 | (45.5) | 22,072 | (44.8) | 0.898 | (0.894, 0.902) | -0.99 | **0.839** | 0.007 | (-0.003, 0.017) | 1.440 | **0.151** |

*(Continued)*

**Table 2.** (Continued)

| Site and population characteristic | Survey | (%) | HDSS | (%) | Survey coverage of HDSS | (95% CI*) | z-score | p-value (H₀ vs Hₐ: p>p₀) | Differ-ence | (95% CI) | z-score | p-value |
|---|---|---|---|---|---|---|---|---|---|---|---|---|
| 0 | 2,213 | (11.2) | 728 | (3.3) | 3.040 | - | - | - | 0.079 | (0.061, 0.097) | 6.380 | < **0.001** |
| 1–4 | 3,399 | (17.2) | 3,794 | (17.2) | 0.896 | (0.886, 0.906) | -0.82 | **0.794** | <0.001 | (-0.017, 0.018) | 0.010 | **0.991** |
| 5–14 | 6,392 | (32.3) | 7,511 | (34.0) | 0.851 | (0.843, 0.859) | -14.16 | **1.000** | -0.017 | (-0.031, -0.001) | -2.110 | **0.035** |
| 15–29 | 3,740 | (18.9) | 5,216 | (23.6) | 0.717 | (0.705, 0.729) | -44.06 | **1.000** | -0.047 | (-0.064, -0.030) | -5.340 | < **0.001** |
| 30–49 | 2,344 | (11.9) | 2,907 | (13.2) | 0.806 | (0.792, 0.820) | -16.89 | **1.000** | -0.013 | (-0.031, 0.005) | -1.420 | **0.155** |
| 50–64 | 1,042 | (5.3) | 1,216 | (5.5) | 0.857 | (0.837, 0.877) | -5.00 | **1.000** | -0.002 | (-0.021, 0.016) | -0.250 | **0.804** |
| 65+ | 632 | (3.2) | 700 | (3.2) | 0.903 | (0.881, 0.925) | 0.26 | **0.396** | <0.001 | (-0.019, 0.019) | 0.030 | **0.978** |
| Not stated | 2 | | | | | | | | | | | |
| Female | 23,689 | (54.4) | 27,215 | (55.2) | 0.870 | (0.866, 0.874) | -15.39 | **1.000** | -0.008 | (-0.017, 0.001) | -1.810 | **0.070** |
| 0 | 2,228 | (9.4) | 690 | (2.5) | 3.229 | - | - | - | 0.069 | (0.052, 0.086) | 5.880 | < **0.001** |
| 1–4 | 3,351 | (14.1) | 3,763 | (13.8) | 0.891 | (0.881, 0.901) | -1.84 | **0.967** | 0.003 | (-0.013, 0.019) | 0.360 | **0.715** |
| 5–14 | 6,281 | (26.4) | 7,373 | (27.1) | 0.852 | (0.844, 0.860) | -13.74 | **1.000** | -0.007 | (-0.021, 0.008) | -0.850 | **0.393** |
| 15–29 | 6,046 | (25.5) | 7,404 | (27.2) | 0.817 | (0.808, 0.826) | -23.81 | **1.000** | -0.018 | (-0.033, -0.003) | -2.300 | **0.021** |
| 30–49 | 3,747 | (15.8) | 5,429 | (20.0) | 0.690 | (0.678, 0.702) | -51.58 | **1.000** | -0.042 | (-0.058, -0.026) | -5.100 | < **0.001** |
| 50–64 | 1,394 | (5.9) | 1,739 | (6.4) | 0.802 | (0.783, 0.821) | -13.62 | **1.000** | -0.005 | (-0.022, 0.012) | -0.600 | **0.546** |
| 65+ | 707 | (3.0) | 817 | (3.0) | 0.865 | (0.842, 0.888) | -3.33 | **1.000** | <-0.001 | (-0.017, 0.017) | -0.030 | **0.976** |
| Not stated | 1 | | | | | | | | | | | |
| Not Stated | 19 | (0.04) | | | | | | | | | | |
| Ethnic Group | | | | | | | | | | | | |
| Mandinka | 8,758 | (20.1) | 10,491 | (21.3) | 0.835 | (0.828, 0.842) | -22.19 | **1.000** | -0.012 | (-0.023, -0.001) | -2.040 | **0.041** |
| Fula | 15,017 | (34.5) | 16,650 | (33.8) | 0.902 | (0.897, 0.907) | 0.86 | **0.195** | 0.007 | (-0.003, 0.017) | 1.310 | **0.190** |
| Sarahule | 17,610 | (40.4) | 20,893 | (42.4) | 0.843 | (0.838, 0.848) | -27.46 | **1.000** | -0.020 | (-0.030, -0.010) | -3.970 | < **0.001** |
| Other | 2,112 | (4.9) | 1,253 | (2.5) | 1.686 | - | - | - | 0.024 | (0.011, 0.037) | 3.430 | **0.001** |
| Not stated | 41 | (0.1) | | | | | | | | | | |

* At a hypothesized proportion of 0.9.

represented the FHDSS in terms of the population count ($p<0.001$). Similar evidence was obtained for the population characteristics of sex and ethnic group, except for Wollof group, for who the survey's coverage of the HDSS was 0.896 (95% CI: 0.890–0.902) and fell just short of the desired threshold.

In Basse, the selected households accounted for a total population of 49,287 in the HDSS and 43,538 covered by the survey, giving a coverage of 0.883 (95% CI: 0.880–0.886). This is less than the hypothesised proportion and the one-sample proportion test confirmed that the household survey in Basse yielded an undercount of the BHDSS in terms of population size ($p = 1$). All population sub-groups in the HDSS were inadequately represented by the survey according to the derived statistical evidence.

In terms of age structure by sex, the survey reported more than double the number of infants (i.e. <1 year) for both sexes than the HDSS in Farafenni; and more than three times for the same age group in Basse. Due to enumerator error for not confirming the date of birth field in the electronic data capture form that was defaulted to current date, dates of birth of many individuals were reported to be similar to the date of interview, thus causing the unexpected excess number of infants aged less than 1 year. Apart from this anomaly, the survey significantly undercounted all defined age groups of the population for both sexes in Basse. In Farafenni, however, only the age groups 15–29 and 30–49 for males, and 30–49 and 65+ for females manifested statistically significant evidence of undercounts.

The two-sample proportion tests investigating revealed statistically significant differences in sample proportions at the 0.05 level for the Wollof ethnic group in Farafenni ($p = 0.003$); and Mandinka and Sarahule groups in Basse ($p = 0.041$ and $<0.001$, respectively). The sex distributions in both data sources were similar at both sites despite the overall undercount reported for Basse. However, structural differences were observed for the age group 15–29 among males ($p = 0.008$) and 30–49 among females ($p = 0.004$) in Farafenni; and the two age groups 5–14 and 15–29 among males ($p = 0.035$ and $<0.001$, respectively), and 15–29 and 30–49 among females ($p = 0.021$ and $<0.001$, respectively) in Basse.

## Comparison of childhood mortality indicators

The pregnancy histories yielded 12,528 live births from 6,239 women aged 15–49 in Farafenni; and 11,106 live births from 9,793 women of the same age group in Basse. The corresponding number of women aged 15–49 in the comparative HDSS datasets were 7,168 and 12,833 respectively. The childhood mortality indicators and rate ratios derived from HDSS and household survey data are presented in Table 3 by site and 5-year periods. In Farafenni, the survey-derived estimates of neonatal mortality were consistently higher than HDSS-based rates throughout the 15-year period. A similar trend was observed for infant mortality estimates, save that the rates in the earliest period were not significantly different from each other with a rate ratio of 1.23 (95% CI: 0.93–1.63). Estimates of child mortality rates from the two data sources were statistically similar for all three periods. In the case of Basse, there were no statistically significant differences between survey and HDSS derived childhood mortality indicators in the most recent 5-year period, which is the only period fully covered by the two data collection methods.

Similar childhood mortality estimates and rate ratios by site and calendar year are presented in Table 4 for the most recent 5-year period, 2011–2015. Whilst a similar trend of consistently higher neonatal mortality estimate from the household survey was observed in Farafenni, the reported rate ratios were statistically significant only for 2013 and 2015, i.e. 2.49 (95% CI: 1.27–4.83) and 6.76 (95% CI: 3.33–13.70), respectively. The relatively high neonatal mortality rate of 63.5 (95% CI: 49.2–81.7) obtained from the survey for Farafenni in 2015 also inflated the corresponding estimates for infant and under-5 mortality. It was only in 2015 that Farafenni recorded statistically different under-5 mortality rates between the two sources of data. In Basse, both data sources produced statistically similar childhood mortality indicators for

**Table 3. Comparison of survey-derived and HDSS-based childhood mortality indicators and rate ratios by site and five-year period, 2001–2005 to 2011–2015.**

| | 2001–2005 | | 2006–2010[‡] | | 2011–2015 | |
|---|---|---|---|---|---|---|
| | Rate/Ratio | (95% CI) | Rate/Ratio | (95% CI) | Rate/Ratio | (95% CI) |
| **FARAFENNI** | | | | | | |
| *Neonatal mortality*[*] | | | | | | |
| Survey | 27.7 | (21.7, 35.2) | 26.4 | (21.6, 32.4) | 31.8 | (26.8, 37.7) |
| HDSS | 17.0 | (12.7, 22.9) | 10.0 | (7.4, 13.6) | 13.9 | (11.0, 17.7) |
| Rate ratio | 1.62 | (1.10, 2.37) | 2.70 | (1.86, 3.91) | 2.29 | (1.75, 3.18) |
| $\chi^2$ | 6.11 | | 29.53 | | 34.07 | |
| *p-value* | *0.0134* | | *<0.0001* | | *<0.0001* | |
| *Infant mortality*[*] | | | | | | |
| Survey | 45.0 | (37.3, 54.4) | 36.4 | (30.6, 43.3) | 41.3 | (35.6, 48.0) |
| HDSS | 37.8 | (31.1, 45.9) | 22.1 | (18.0, 27.1) | 25.1 | (21.0, 29.9) |
| Rate ratio | 1.23 | (0.93, 1.63) | 1.71 | (1.30, 2.25) | 1.73 | (1.37, 2.19) |
| $\chi^2$ | 2.18 | | 15.22 | | 21.20 | |
| *p-value* | *0.1396* | | *0.0001* | | *<0.0001* | |
| *Child mortality*[†] | | | | | | |
| Survey | 30.1 | (23.3, 38.9) | 13.1 | (9.4, 18.1) | 14.6 | (11.1, 19.0) |
| HDSS | 39.2 | (32.4, 47.4) | 19.2 | (15.3, 24.0) | 17.4 | (14.1, 21.6) |
| Rate ratio | 0.78 | (0.56, 1.08) | 0.70 | (0.47, 1.04) | 0.85 | (0.60, 1.19) |
| $\chi^2$ | 2.30 | | 3.10 | | 0.91 | |
| *p-value* | *0.1290* | | *0.0782* | | *0.3401* | |
| *Under-5 mortality*[*] | | | | | | |
| Survey | 73.8 | (63.5, 85.7) | 49.0 | (42.1, 57.1) | 55.3 | (48.6, 62.9) |
| HDSS | 75.5 | (66.1, 86.3) | 40.9 | (35.2, 47.5) | 42.0 | (36.7, 48.1) |
| Rate ratio | 1.08 | (0.88, 1.33) | 1.37 | (1.10, 1.70) | 1.40 | (1.16, 1.70) |
| $\chi^2$ | 0.49 | | 7.87 | | 12.10 | |
| *p-value* | *0.4843* | | *0.0050* | | *0.0005* | |
| **BASSE** | | | | | | |
| *Neonatal mortality*[*] | | | | | | |
| Survey | 28.8 | (22.4, 36.9) | 18.2 | (13.9, 23.6) | 17.9 | (14.1, 22.8) |
| HDSS | | | 10.9 | (8.7, 13.6) | 16.2 | (13.9, 19.0) |
| Rate ratio | | | 1.67 | (1.18, 2.36) | 1.11 | (0.83, 1.49) |
| $\chi^2$ | | | 8.45 | | 0.53 | |
| *p-value* | | | *0.0036* | | *0.4661* | |
| *Infant mortality*[*] | | | | | | |
| Survey | 55.8 | (46.7, 66.6) | 36.6 | (30.4, 44.1) | 29.1 | (24.1, 35.2) |
| HDSS | | | 29.1 | (25.5, 33.3) | 32.5 | (29.1, 36.2) |
| Rate ratio | | | 1.29 | (1.02, 1.63) | 0.92 | (0.74, 1.15) |
| $\chi^2$ | | | 4.70 | | 0.55 | |
| *p-value* | | | *0.0302* | | *0.4583* | |
| *Child mortality*[†] | | | | | | |
| Survey | 42.5 | (33.8, 53.2) | 24.2 | (18.7, 31.2) | 22.3 | (17.7, 27.9) |
| HDSS | | | 37.3 | (33.1, 42.1) | 27.4 | (24.3, 30.9) |
| Rate ratio | | | 0.66 | (0.50, 0.88) | 0.82 | (0.63, 1.06) |
| $\chi^2$ | | | 8.30 | | 2.39 | |
| *p-value* | | | *0.0040* | | *0.1222* | |
| *Under-5 mortality*[*] | | | | | | |
| Survey | 95.9 | (83.6, 110) | 59.9 | (51.6, 69.5) | 50.8 | (44.0, 58.6) |

*(Continued)*

**Table 3.** (Continued)

| | 2001–2005 | | 2006–2010[‡] | | 2011–2015 | |
|---|---|---|---|---|---|---|
| | Rate/Ratio | (95% CI) | Rate/Ratio | (95% CI) | Rate/Ratio | (95% CI) |
| HDSS | | | 65.4 | (59.8, 71.4) | 59.0 | (54.5, 63.9) |
| Rate ratio | | | 1.00 | (0.84, 1.20) | 0.89 | (0.75, 1.06) |
| $\chi^2$ | | | 0.00 | | 1.75 | |
| *p-value* | | | *0.9626* | | *0.1861* | |

[*] *Rates per 1,000 live births.*

[†] *Rates per 1,000 population aged 1 year.*

[‡]*The Basse HDSS covered July 2007 –December 2010 of this period.*

each year, except for child mortality in 2012 (RR = 0.47; 95% CI: 0.23–0.95) and under-5 mortality in 2011 (RR = 0.64; 95% CI: 0.42–0.97).

## Discussion

This study attempts to ascertain whether a DHS-type cross-sectional household survey can be used to assess the quality of data and reliability of a prospective demographic surveillance system. The focus in this attempt is limited to determining if the two different methods of data collection applied to the same population yield similar measures in terms of size, structure and commonly derivable childhood mortality indicators. These methods are based on distinctly different operational definitions, and have contrasting advantages and disadvantages associated with them by virtue of their designs. Such are their basic theoretical differences that, even if reporting and data collection were perfect, they would not yield the same datasets. Due consideration must therefore be taken of the key features of each method in the comparison of data independently generated by them with a view to ascertaining whether they adequately match each other, with one hence validating the other.

In terms of population size and structure, the adapted DHS-type survey sufficiently validates the HDSS data in Farafenni; whilst it fell short of covering the Basse HDSS assuming, *a priori*, a minimum acceptable coverage proportion of 0.9. This may be related to the fact that household sizes are larger in Basse than in Farafenni. Since the population sizes represent the total listed household members in the survey and the population in the HDSS as at the time of the household survey, differences between the two sources of data can only be attributed to their operational definitions of residency. Because the HDSS rosters are based on periodic visits, they may retain as active the residency status of a household member who is away from the household for up to four months. The cross-sectional household survey, in contrast, is fully up-to-date on the day of the survey but does not capture who was resident the last time the HDSS fieldworker visited the household.

The comparison of household size ranges from the survey against those indicated in the HDSS database shows that 517 (18%) households in Basse were under-enumerated at varying degrees, including 30 households with sizes in excess of 60 but reported to have only up to 40 members in the survey (Table 1). On the other hand, 238 (8%) were over-enumerated, but at lesser extent than those that were undercounted. This is evidently a contributory factor to the observed survey undercount in Basse. Some survey respondents misinterpreted the definition of a household provided by the enumerators to mean their immediate nuclear family and thereby listed far fewer members of the household than documented in the HDSS database. The sex structures from the two data sources in Basse also reveal that the undercount affected

**Table 4. Comparison of survey-derived and HDSS-based annual childhood mortality indicators and rate ratios by site, 2011–2015.**

| | 2011 | | 2012 | | 2013 | | 2014 | | 2015 | |
|---|---|---|---|---|---|---|---|---|---|---|
| | | *(95% CI)* | | *(95% CI)* | | *(95% CI)* | | *(95% CI)* | | *(95% CI)* |
| **FARAFENNI** | | | | | | | | | | |
| *Neonatal mortality* | | | | | | | | | | |
| Survey | 21.5 | *(13.3, 34.9)* | 18.9 | *(11.4, 31.1)* | 31.3 | *(21.3, 46.0)* | 19.3 | *(11.7, 31.8)* | 63.5 | *(49.2, 81.7)* |
| HDSS | 11.4 | *(6.1, 21.1)* | 17.7 | *(11.4, 27.7)* | 13.0 | *(7.6, 22.4)* | 15.7 | *(9.5, 26.0)* | 10.7 | *(5.6, 20.5)* |
| Rate ratio | 1.90 | *(0.86, 4.20)* | 1.06 | *(0.54, 2.09)* | 2.47 | *(1.27, 4.83)* | 1.23 | *(0.60, 2.51)* | 6.74 | *(3.34, 13.60)* |
| $\chi^2$ | 2.65 | | 0.03 | | 7.50 | | 0.32 | | 37.92 | |
| *p-value* | *0.1038* | | *0.8620* | | *0.0062* | | *0.5741* | | *<0.0001* | |
| *Infant mortality* | | | | | | | | | | |
| Survey | 32.2 | *(21.7, 47.6)* | 26.7 | *(17.5, 40.7)* | 41.4 | *(29.6, 57.7)* | 29.7 | *(19.9, 44.4)* | 72.1 | *(56.7, 91.3)* |
| HDSS | 18.3 | *(11.3, 29.8)* | 31.7 | *(22.6, 44.3)* | 23.4 | *(15.8, 34.8)* | 28.0 | *(19.3, 40.6)* | 22.2 | *(14.4, 34.2)* |
| Rate ratio | 1.74 | *(0.93, 3.28)* | 0.83 | *(0.48, 1.44)* | 1.85 | *(1.09, 3.13)* | 1.09 | *(0.63, 1.90)* | 3.84 | *(2.32, 6.34)* |
| $\chi^2$ | 3.04 | | 0.43 | | 5.43 | | 0.09 | | 31.83 | |
| *p-value* | *0.0811* | | *0.5112* | | *0.0198* | | *0.7611* | | *<0.0001* | |
| *Child mortality* | | | | | | | | | | |
| Survey | 19.5 | *(11.3, 33.5)* | 16.5 | *(9.4, 28.8)* | 8.4 | *(3.8, 18.6)* | 9.1 | *(4.4, 19.0)* | 20.1 | *(12.2, 33.1)* |
| HDSS | 20.4 | *(12.9, 32.3)* | 15.4 | *(9.1, 25.8)* | 21.6 | *(14.1, 32.9)* | 14.9 | *(9.0, 24.5)* | 15.1 | *(9.1, 24.9)* |
| Rate ratio | 0.95 | *(0.46, 1.93)* | 1.11 | *(0.51, 2.39)* | 0.38 | *(0.15, 0.93)* | 0.63 | *(0.26, 1.55)* | 1.36 | *(0.66, 2.77)* |
| $\chi^2$ | 0.02 | | 0.07 | | 4.87 | | 1.03 | | 0.70 | |
| *p-value* | *0.8761* | | *0.7952* | | *0.0274* | | *0.3096* | | *0.4035* | |
| *Under-5 mortality* | | | | | | | | | | |
| Survey | 51.1 | *(37.2, 69.9)* | 42.7 | *(30.6, 59.6)* | 49.5 | *(36.4, 67.1)* | 38.6 | *(27.1, 54.7)* | 90.7 | *(73.3, 112.0)* |
| HDSS | 38.4 | *(27.6, 53.4)* | 46.6 | *(35.2, 61.6)* | 44.5 | *(33.4, 59.2)* | 42.5 | *(31.5, 57.0)* | 36.9 | *(26.6, 51.1)* |
| Rate ratio | 1.39 | *(0.87, 2.21)* | 0.91 | *(0.58, 1.42)* | 1.14 | *(0.74, 1.75)* | 0.96 | *(0.60, 1.53)* | 2.96 | *(1.98, 4.41)* |
| $\chi^2$ | 1.93 | | 0.17 | | 0.37 | | 0.04 | | 31.28 | |
| *p-value* | *0.1645* | | *0.6765* | | *0.5403* | | *0.8473* | | *<0.0001* | |
| **BASSE** | | | | | | | | | | |
| *Neonatal mortality* | | | | | | | | | | |
| Survey | 11.8 | *(5.9, 23.5)* | 14.6 | *(8.1, 26.3)* | 19.4 | *(11.6, 32.6)* | 27.1 | *(17.5, 41.6)* | 16.3 | *(9.3, 28.6)* |
| HDSS | 16.5 | *(11.5, 23.5)* | 14.6 | *(10.2, 21)* | 14.0 | *(9.7, 20.2)* | 18.7 | *(13.5, 25.8)* | 17.8 | *(12.4, 25.5)* |
| Rate ratio | 0.72 | *(0.33, 1.56)* | 1.00 | *(0.50, 2.00)* | 1.42 | *(0.75, 2.69)* | 1.46 | *(0.84, 2.52)* | 0.94 | *(0.48, 1.83)* |
| $\chi^2$ | 0.70 | | 0.00 | | 1.15 | | 1.84 | | 0.04 | |
| *p-value* | *0.4012* | | *0.9975* | | *0.2846* | | *0.1754* | | *0.8443* | |
| *Infant mortality* | | | | | | | | | | |
| Survey | 20.7 | *(12.3, 34.7)* | 27.5 | *(17.8, 42.3)* | 29.1 | *(19.1, 44.3)* | 38.3 | *(26.6, 54.9)* | 29.8 | *(19.7, 44.9)* |
| HDSS | 35.8 | *(28.3, 45.3)* | 31.9 | *(25, 40.6)* | 32.5 | *(25.6, 41.1)* | 31.6 | *(24.7, 40.4)* | 30.4 | *(23.3, 39.7)* |
| Rate ratio | 0.58 | *(0.33, 1.04)* | 0.88 | *(0.53, 1.45)* | 0.89 | *(0.55, 1.46)* | 1.24 | *(0.79, 1.94)* | 0.98 | *(0.64, 1.73)* |
| $\chi^2$ | 3.47 | | 0.27 | | 0.20 | | 0.88 | | 0.04 | |
| *p-value* | *0.0623* | | *0.6014* | | *0.6558* | | *0.3493* | | *0.8412* | |
| *Child mortality* | | | | | | | | | | |
| Survey | 22.4 | *(13.0, 38.3)* | 13.9 | *(7.3, 26.6)* | 29.8 | *(19.3, 45.9)* | 14.3 | *(7.7, 26.4)* | 30.1 | *(19.7, 45.8)* |
| HDSS | 32.8 | *(25.5, 42.1)* | 30.6 | *(23.7, 39.5)* | 31.0 | *(24.1, 39.9)* | 24.5 | *(18.5, 32.3)* | 18.9 | *(13.6, 26)* |
| Rate ratio | 0.67 | *(0.37, 1.23)* | 0.47 | *(0.23, 0.95)* | 0.97 | *(0.58, 1.61)* | 0.59 | *(0.30, 1.16)* | 1.62 | *(0.95, 2.78)* |
| $\chi^2$ | 1.68 | | 4.65 | | 0.02 | | 2.37 | | 3.14 | |
| *p-value* | *0.1949* | | *0.0311* | | *0.8971* | | *0.1235* | | *0.0762* | |
| *Under-5 mortality* | | | | | | | | | | |
| Survey | 42.6 | *(29.4, 61.7)* | 41.0 | *(28.7, 58.5)* | 58.0 | *(43.0, 78.1)* | 52.0 | *(38.1, 70.8)* | 59.0 | *(44.1, 78.8)* |

*(Continued)*

**Table 4.** (Continued)

| | 2011 | | 2012 | | 2013 | | 2014 | | 2015 | |
|---|---|---|---|---|---|---|---|---|---|---|
| | | *(95% CI)* | | *(95% CI)* | | *(95% CI)* | | *(95% CI)* | | *(95% CI)* |
| HDSS | 67.4 | *(56.9, 79.8)* | 61.5 | *(51.7, 73.1)* | 62.5 | *(52.7, 74.0)* | 55.3 | *(46.0, 66.3)* | 48.7 | *(39.7, 59.7)* |
| Rate ratio | 0.64 | *(0.42, 0.97)* | 0.71 | *(0.47, 1.06)* | 0.93 | *(0.66, 1.33)* | 0.98 | *(0.68, 1.41)* | 1.32 | *(0.92, 1.89)* |
| $\chi^2$ | 4.43 | | 2.83 | | 0.15 | | 0.02 | | 2.21 | |
| *p-value* | *0.0353* | | *0.0926* | | *0.6989* | | *0.8987* | | *0.1369* | |

males more than females (Table 1). This is expected as a considerable proportion of men in this population are engaged in trade and more likely to travel outside the surveillance area than women.

The designs of the two data collection strategies also impact on estimates of childhood mortality indicators differently. As in any DHS, pregnancy histories are characterised by recall bias whose magnitude increases with time since the termination of the pregnancy [23, 24]. Also, mis-statement of dates of birth and death of children by mothers can cause displacement within the respective age categories considered in the analysis, as well as across calendar periods [25]. For the prospective demographic surveillance systems, pregnancies and their outcomes, in particular pregnancies that end in early neonatal deaths, can be missed between rounds of data collection [26]. The result in Farafenni, therefore, that consistently higher neonatal mortality rates were obtained from the household survey than were derived from the HDSS data for the 15-year period considered in the analysis is not particularly surprising. It has been observed in another comparative study of these HDSS data and the first Gambian DHS data [27], as well as at other HDSS sites [10]. Notwithstanding the higher neonatal mortality rates from the household survey, the derived infant and under-5 mortality rates for the earlier period of 2001–2005 were statistically similar to those obtained from the HDSS data. The observed statistically significant differences in these rates for the two most recent periods can be attributed to two factors, each affecting one period. The first factor is the disruption of surveillance in Farafenni for 13 months between March 2008 and April 2009, which inevitably caused a significant number of neonatal deaths to be missed between 2008 and 2010, hence the low rate of 10.0 per 1,000 live births recorded for the period 2006–2010. The impact of this break is clearly demonstrated in the annual trends in neonatal, infant and child mortality rates from both data sources presented in Fig 2.

For the most recent period 2011–2015, a relatively high level of neonatal mortality was obtained from the survey for Farafenni, i.e. 63.5 per 1,000 live births compared to 10.7 per 1,000 live births from the HDSS. According to the pregnancy history data, it is the highest level of annual neonatal mortality rate ever recorded in Farafenni. However, in a community reported to have attained its MDG4 goal seven years before time [1], the reported level of neonatal mortality for 2015 in Farafenni is likely to be due to displacement error of births and deaths of neonates caused either by wrong reports by mothers or entry errors on the part of enumerators. The situation in Basse is completely different in that almost all annual childhood mortality indicators from both data sources are statistically similar (Table 4). This is further demonstrated in Fig 2.

Although the survey in Basse fell short of providing an acceptable coverage of the HDSS based on a hypothesised proportion of 0.9, the pregnancy histories have provided childhood mortality rates similar to those obtained from the HDSS for the period 2011–2015. Since it was shown that men were more likely to be missed in such surveys, and simulations have demonstrated that HDSS data relating to population structure and derivable mortality rates can

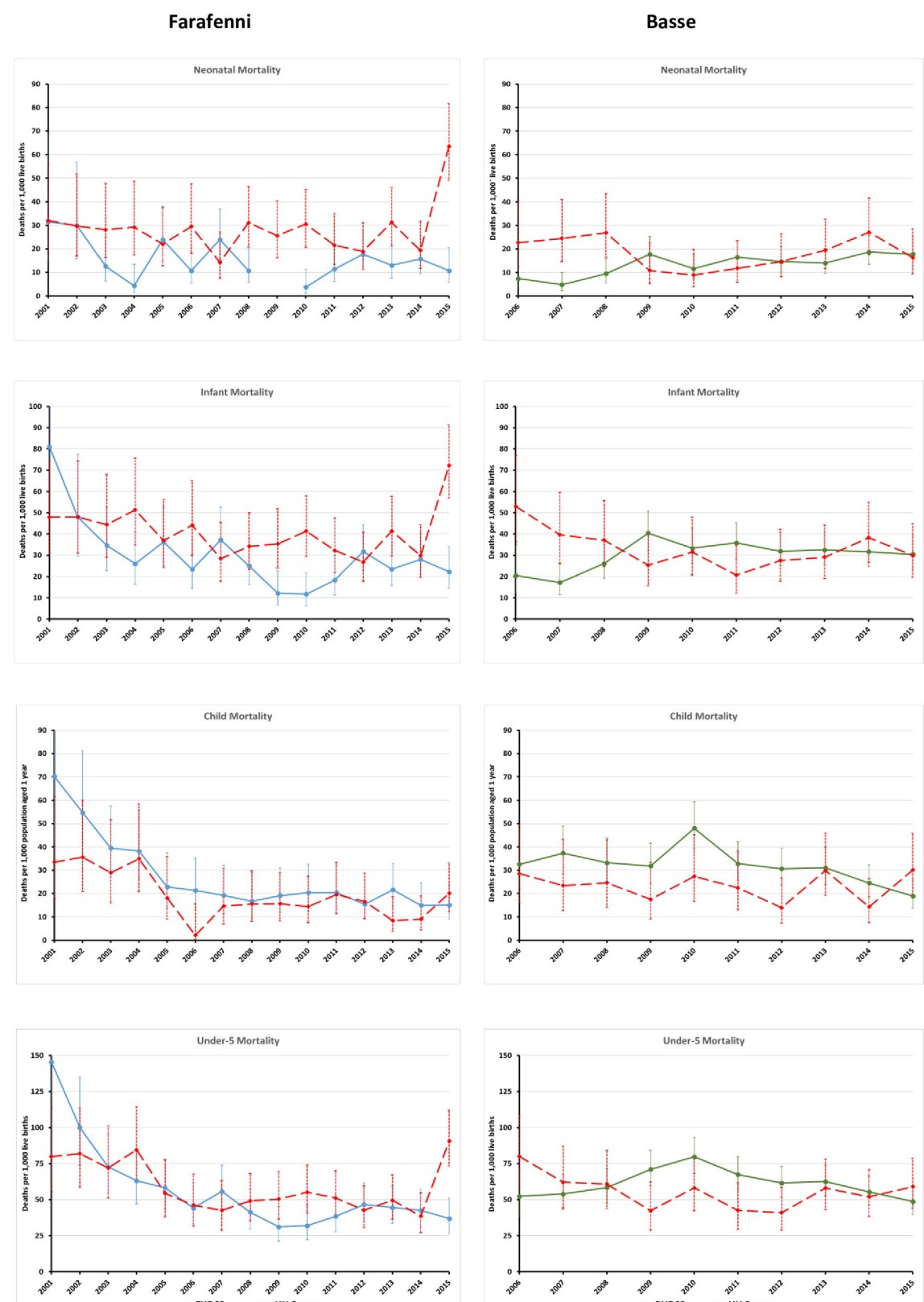

**Fig 2. Comparison of trends in survey-derived and HDSS-based childhood mortality indicators by site, Farafenni, 2001–2015, and Basse, 2006–2015.**

withstand random errors of up to 20% [6], it is possible to reconsider lowering the assumed *a priori* hypothesised proportion to 0.85. If such a level were used in the analysis presented in this study, the household surveys in both Farafenni and Basse would have sufficiently represented the respective HDSS data in terms of population size and structure, and derived childhood mortality indicators would remain the same.

Whilst the results clearly demonstrate that an adapted DHS-type survey can be used to validate routine HDSS data using population structure and childhood mortality, there are other aspects of HDSS data that can be validated in a similar manner, such as estimates fertility. Ideally, we would have liked to compare the two data sources in terms of whether they identified similar groups of women, and whether the women reported similar childbearing histories, and whether the two sets of children have similar levels and trends of mortality. However, this extended analysis could not be undertaken due to the poor quality of date-of-birth information collected by the household survey. The field for date of birth in the electronic survey questionnaire was designed to default to "current date" or date of interview and to be set accordingly by the enumerator depending on the reported date of birth to be followed by the return key for confirmation of the date entered. However, due to enumerator error, entry of reported date of birth was not confirmed in many cases using the return key, hence resulting in dates of birth being the same as interview dates. This is responsible for the significantly exaggerated proportions aged 0 in both populations and for both sexes (see Table 2).

Considering the high cost of conducting such adapted DHS-type household surveys in a setting as resource intensive as the HDSS, further research will be required to establish the minimum representative sample size that will suffice to efficiently validate or assess HDSS data quality and database integrity on a regular basis. Whilst this study was based on sites with similar update round intervals of four months, similar trials of the method should be made at sites with update round intervals of six months and one year to ascertain the impact of data collection intervals on the suitability of the adapted DHS survey to assess HDSS data.

## Supporting information

**S1 File.**
(PDF)

**S2 File.**
(PDF)

**S3 File.**
(PDF)

**S4 File.**
(PDF)

**S5 File.**
(DOCX)

## Acknowledgments

We are grateful to the two teams of female enumerators who conducted the household surveys in Farafenni and Basse. We also thank Pierre Gomez and Mamadi Sidibeh for the support and guidance they accorded to the enumerators, especially in the identification of households.

## Author Contributions

**Conceptualization:** Momodou Jasseh.

**Data curation:** Momodou Jasseh.

**Formal analysis:** Momodou Jasseh.

**Investigation:** Anne J. Rerimoi.

**Methodology:** Momodou Jasseh, Ian M. Timæus.

**Project administration:** Anne J. Rerimoi.

**Supervision:** Anne J. Rerimoi.

**Validation:** Momodou Jasseh, Georges Reniers, Ian M. Timæus.

**Writing – original draft:** Momodou Jasseh.

**Writing – review & editing:** Momodou Jasseh, Anne J. Rerimoi, Georges Reniers, Ian M. Timæus.

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
