## [Decision Letter · Decision Letter 0]

9 Mar 2022

PONE-D-21-37201Assessment of the consistency of health and demographic surveillance and household survey data: A demonstration at two HDSS sites in The Gambia.PLOS ONE

Dear Dr. Jasseh,

Thank you for submitting your manuscript to PLOS ONE. After careful consideration, we feel that it has merit but does not fully meet PLOS ONE’s publication criteria as it currently stands. Therefore, we invite you to submit a revised version of the manuscript that addresses the points raised during the review process.

We look forward to receiving your revised manuscript.

Kind regards,

Orvalho Augusto, MD, MPH

Academic Editor

PLOS ONE

Journal Requirements:

The Farafenni and Basse HDSS sites are supported by Medical Research Council Unit The Gambia at London School of Hygiene and Tropical Medicine. The household survey was supported by a grant from the University of Nagasaki, Japan. 

The Farafenni and Basse HDSS sites are supported by Medical Research Council Unit The Gambia at London School of Hygiene and Tropical Medicine. The household survey was supported by a grant from the University of Nagasaki, Japan. The funders had no role in study design, data collection and analysis, decision to publish, or preparation of the manuscript.

Additional Editor Comments:

This is a very important report on the design and methodology challenges of childhood mortality in a low middle-income country. The authors leverage their 2 Health Demographic Surveillance System (HDSS) capturing births, deaths and migration for at least a decade. On these sites, they applied a DHS-like survey for around ~2500 households and collect full birth history for women between the ages 15 to 49 years old. Population enumeration and childhood mortality (neonatal, infant and child mortality) are compared through proportion coverage, proportion differences and rate ratios.

Issues:

1. Please add more information some demographic information of these sites: eg: Education level, and patterns of migration.

2. Provide some information about the enumerators/interviewers (training, schooling level and ages).

3. The childhood mortality here is reported as neonatal, infant, child and under-5 mortality rates. These measures although standard there is still room for confusion. Please add a definition for each somewhere in the methods section.

4. Statistical methods:

- The authors used the Kaplan-Meier method to estimate the mortality rates. This is fine. However, the DHS surveys use discrete survival analysis by computing monthly mortality probabilities. Then taking cumulative survival for the periods of interest. This analysis can be presented in the supplementary materials.

- A critical advantage that the authors throw away is the household pairing on HDSS and on the survey. For example, the exact number of discrepancies on household size could be computed (Table 1) rather than, first, building categories of household sizes and then doing a crosstab. As a result differences within 10 units are considered 0.

- Moreover, the whole analysis happens as if the independence assumption is fulfilled. I am OK with the current procedure but the authors should be aware that we have matched households here and perhaps this should be discussed.

5. Can you provide some description of the respondents? Are they differences between those who respond to the HDSS and those who responded to the DHS-like survey for the enumeration of the household members?

6. The date of birth issue on the survey should be reported in the results. Not too late as just one element of the discussion.

7. For all tables please change the separator of 95%CI. The dash causes confusion with the negative numbers on differences in table 2.

8. Table 3 and Table 4:

- Some of the rate ratios are not replicable. For example among the period 2011-2015 the neonatal mortality rate ratio 31.8/13.9 = 2.29 not the current 2.36

- Please add on the footnote how the chi-squared was computed.

Reviewers' comments:

Reviewer's Responses to Questions

**Comments to the Author**

1. Is the manuscript technically sound, and do the data support the conclusions?

Reviewer #1: Yes

Reviewer #2: No

2. Has the statistical analysis been performed appropriately and rigorously? 

Reviewer #1: Yes

Reviewer #2: Yes

3. Have the authors made all data underlying the findings in their manuscript fully available?

Reviewer #1: Yes

Reviewer #2: Yes

4. Is the manuscript presented in an intelligible fashion and written in standard English?

Reviewer #1: Yes

Reviewer #2: Yes

5. Review Comments to the Author

Reviewer #1: This is a well-written manuscript for a well-designed study to assess the data quality of HDSS in two sites in the Gambia. The assessment adopted a DHS-like household surveys. I do not have specific comments on the contents of the manuscript since it is good as it is. I would just recommend if the manuscript can be reviewed by an editor to check the flow and for any typos.

Reviewer #2: I have identified several technical problems and one research design problem. The technical problems are as follows:

- The tests were conducted in a very conventional way, with the null hypothesis H0 strictly defined as Ps = Ph. However, one could allow a margin of, say, 5% around each estimate. Would a difference of 5% or 10% still be considered a significant difference? The authors could consider using the concept of Smallest effect size of interest (SESOI; see articles by Daniël Lakens). Without such a cut-off point, any small p-value with large samples could be overly interpreted as significant, and large p-values with small samples could be overly interpreted as no difference. In other words, taking the CI into account in addition to the p-value could lead to more nuanced results.

- I did not understand the explanation (lines 263-264, p 11) about the specific annual analysis conducted on the "most recent-5-year period, which is the only period fully covered by the two data collection methods". Table 2 and Figure 2 provide results from both data sources over the period 2001-2015 (Farafenni) or the period 2006-2015 (Basse), so why focus on the period 2011-2015? Accuracy of data? Avoiding recall bias?

- The reader is led to believe that the HDSS data collection misses a number of children, but then the reader discovers a huge bias in the survey data. At the very end of the discussion, a very important methodological limitation of the survey is pointed out (lines 349-350; p14): "the poor quality of date-of-birth information collected by the household survey", leading to "the significantly exaggerated propotions aged 0" in both sites. Why were these cases not cleaned up before analysis? How were these survey records treated in the mortality analysis?

The question of research design is related to the very purpose of the analysis. Mothers' IDs were matched between the 2 sources but not children's IDs (lines 174-175, p 7): this is a serious limitation of the comparison exercise, preventing the identification of children who were missed, or events that were displaced, in either source. In fact, neither the survey nor the HDSS can be considered the most reliable source. However, the authors want to test whether the survey can be used to assess the data quality and reliability of the HDSS data (lines 278-279: p 12). To this question, given the limitations of the survey and the lack of matching of children, the answer is clearly no, even before any comparative analysis is done. Thus, I disagree with the conclusion (lines 344-345; p14) that "the results clearly demonstrate that an adapted DHS-type survey can be used to validate routine HDSS data". The authors mention another, more nuanced objective, which is to check whether the two sources "adequately match each other, with one hence validating the other" (lines 286-287; p12), but I doubt that they can even achieve this objective. I am fairly confident that some of the difference between the two sources is explained by operational definitions of residence and by missed and misplaced events due to the pitfalls of the two data collection procedures, but without a more rigorous comparison it is very difficult to determine which data collection bias leads to which bias in the mortality indicators.

6. PLOS authors have the option to publish the peer review history of their article (what does this mean?). If published, this will include your full peer review and any attached files.

Reviewer #1: No

Reviewer #2: No

---

## [Author Response · Author response to Decision Letter 0]

2 Jun 2022

Editor’s Comments

1. Please add more information some demographic information of these sites: e.g.: Education level, and patterns of migration.

Additional demographic information on the sites have been included under “Study areas and populations” in the Methods section (lines 114-120) and two additional references cited, i.e. Refs 15 and 16 as follows:

• Gambia Bureau of Statistics (GBoS), ICF. The Gambia Demographic and Health Survey 2019-20. Banjul, The Gambia and Rockville, Maryland, USA: GBoS and ICF; 2021.

• Gambia Bureau of Statistics. 2013 Population and Housing Census Report, Volume 4: National Migration Analysis. Banjul, The Gambia: Gambia Bureau of Statistics; 2015.

2. Provide some information about the enumerators/interviewers (training, schooling level and ages).

These have been inserted in lines 126-127 and 143-144 in the revised manuscript with tracked changes.

3. The childhood mortality here is reported as neonatal, infant, child and under-5 mortality rates. These measures although standard there is still room for confusion. Please add a definition for each somewhere in the methods section.

The definitions of the childhood mortality indicators have been adjusted in the last paragraph under “Data and Statistical Analysis” in the Methods section, to enhance clarity.

4. Statistical methods:

- The authors used the Kaplan-Meier method to estimate the mortality rates. This is fine. However, the DHS surveys use discrete survival analysis by computing monthly mortality probabilities. Then taking cumulative survival for the periods of interest. This analysis can be presented in the supplementary materials.

It is correct that actual DHS surveys use discrete survival analysis because of the adoption of the century month calendar (CMC) as standard exposure time capture. However, in the case of our DHS-type household survey, exact calendar dates were used, and similar to the HDSS, thereby enabling us to use the K-M method in the analysis of both datasets. 

The Stata output of the analysis is included as supplementary material.

- A critical advantage that the authors throw away is the household pairing on HDSS and on the survey. For example, the exact number of discrepancies on household size could be computed (Table 1) rather than, first, building categories of household sizes and then doing a crosstab. As a result differences within 10 units are considered 0.

This query describes exactly what we did to generate Table 1. The exact differences between HDSS and reported survey household sizes were determined first before grouping them in categories and cross-tabulating them to produce Table 1 with the main purpose of demonstrating that large HDSS households, usually comprising of big extended families, are more likely to be undercounted in the survey as is the case in Basse. This arises when the respondent decides to respond on behalf of his/her immediate family following the operational definition of “household” read out by the enumerator, rather than reporting for the entire household as registered in the HDSS. 

As the Editor rightly pointed, differences of up to 9 persons between HDSS and survey household sizes may fall within the same category, but that is within the reasonable expected range of discrepancies on the basis of the differences of the two data collection methods adopted.

- Moreover, the whole analysis happens as if the independence assumption is fulfilled. I am OK with the current procedure but the authors should be aware that we have matched households here and perhaps this should be discussed.

It is not clear what ‘independence assumption’ is being referred to here. However, we have made it clear in the manuscript that the HDSS and Household Survey data are completely independent of each other but refer to the same households/population; and different teams of enumerators were engaged in each data collection strategy. All the two datasets have in common is the randomly selected household identification numbers (HHIDs), which were extracted from the HDSS database to enable the Household Survey team locate the sampled households. If this is the ‘independence assumption’ being alluded to, then yes, it is fulfilled.

5. Can you provide some description of the respondents? Are they differences between those who respond to the HDSS and those who responded to the DHS-like survey for the enumeration of the household members?

In both data collection methods, the household heads, or their representatives, are the usual respondents for general household questions; and other household members are called upon for more specific information such as pregnancy status, outcomes, etc. In the DHS-type household survey, the woman’s questionnaire was directly administered for every eligible woman confirmed to be resident in the household by the household head. These points are clarified in the revised manuscript in lines 132-136, and 159-160.

6. The date of birth issue on the survey should be reported in the results. Not too late as just one element of the discussion.

The issue has been reflected in the Results section as suggested (lines 255-258).

7. For all tables please change the separator of 95%CI. The dash causes confusion with the negative numbers on differences in table 2.

The dashes have been replaced with a comma separator in all the tables.

8. Table 3 and Table 4:

- Some of the rate ratios are not replicable. For example among the period 2011-2015 the neonatal mortality rate ratio 31.8/13.9 = 2.29 not the current 2.36

- Please add on the footnote how the chi-squared was computed.

The original State scripts were re-run using Stata 17. This is adjusted accordingly in Ref 19 in the revised manuscript. Also, all resulting minor decimal changes in the results have been updated in Tables 3 and 4. It is interesting to note that whilst the reported details for the neonatal mortality rate ratio is 2.29 when computed directly (and corrected in the revised manuscript), the Stata output still returns a rate ratio of 2.36 and all other details remain the same. The reported chi-squared were generated by Stata as reported in the accompanying supplementary material.

Reviewer #1 Comments: 

This is a well-written manuscript for a well-designed study to assess the data quality of HDSS in two sites in the Gambia. The assessment adopted a DHS-like household surveys. I do not have specific comments on the contents of the manuscript since it is good as it is. I would just recommend if the manuscript can be reviewed by an editor to check the flow and for any typos.

We thank Reviewer #1 for his understanding of our work.

Reviewer #2 Comments: 

-I have identified several technical problems and one research design problem. The technical problems are as follows:

- The tests were conducted in a very conventional way, with the null hypothesis H0 strictly defined as Ps = Ph. However, one could allow a margin of, say, 5% around each estimate. Would a difference of 5% or 10% still be considered a significant difference? The authors could consider using the concept of Smallest effect size of interest (SESOI; see articles by Daniël Lakens). Without such a cut-off point, any small p-value with large samples could be overly interpreted as significant, and large p-values with small samples could be overly interpreted as no difference. In other words, taking the CI into account in addition to the p-value could lead to more nuanced results.

We note with interest Reviewer #2’s suggestion on the adoption of the “smallest effect size of interest” concept. The Lead Author has perused samples of Daniël Lakens’s work as well as watched some of his lectures on YouTube video. However, it is apparent that the context in which the concept of SESOI is used in his work on psychology and psychological science is far more complex than the simple, but appropriate, technique we have adopted to compare two sample proportions, based on an established technique described by Sabo and Boone in their book “Essential Research Methods – A Guide for Non-Statisticians” (Ref 18 in the revised manuscript). We are therefore convinced that the method we have used suffices and is appropriate for the task at hand.

- I did not understand the explanation (lines 263-264, p 11) about the specific annual analysis conducted on the "most recent-5-year period, which is the only period fully covered by the two data collection methods". Table 2 and Figure 2 provide results from both data sources over the period 2001-2015 (Farafenni) or the period 2006-2015 (Basse), so why focus on the period 2011-2015? Accuracy of data? Avoiding recall bias?

The phrase quoted by Reviewer #2 in lines 263-264 in the original submission refer to the comparison of the mortality indicators derived for Basse from the HDSS and the Household Survey for the 5-year period 2011-2015. This is the only period that mortality rates could be estimated from both sources of data for Basse since the HDSS was operational from July 2007 as noted at the foot of Table 3.

Our focus on the most recent 5-year period, i.e. 2011-2015, for estimation of “annual” mortality indicators of interest is to further enable us explore the consistency between the two datasets when they are unpacked from a wider 5-year window to much narrower yearly periods. This is exactly the added value to the analysis we have demonstrated in Table 4. 

- The reader is led to believe that the HDSS data collection misses a number of children, but then the reader discovers a huge bias in the survey data. At the very end of the discussion, a very important methodological limitation of the survey is pointed out (lines 349-350; p14): "the poor quality of date-of-birth information collected by the household survey", leading to "the significantly exaggerated proportions aged 0" in both sites. Why were these cases not cleaned up before analysis? How were these survey records treated in the mortality analysis?

The first part of this query is similar to that raised by the Editor (point 6) and has been addressed accordingly.

As to why the affected cases were not cleaned up before the analysis, we wish to draw the attention of the Reviewer to the following points:

a. That the objective of the study is to establish whether a sample DHS-type cross-sectional household survey conducted on an HDSS platform can be used to assess the quality of it routinely collected data in terms of population counts, characteristics and childhood mortality measures. The suggested cleaning would have meant extracting their DOBs from the HDSS database, thus defeating the purpose of the demonstration of the consistency of the two datasets that the assessment had sought to establish.

b. The household survey did not collect unique IDs for individuals. Therefore, any attempt of extracting DOBs from the HDSS database will be based on identifying individual by names and relationships. Whilst this possible, extremely time consuming and may be necessary for another type of comparison on the two data sources, the process is not required for this particular analysis.

-The question of research design is related to the very purpose of the analysis. Mothers' IDs were matched between the 2 sources but not children's IDs (lines 174-175, p 7): this is a serious limitation of the comparison exercise, preventing the identification of children who were missed, or events that were displaced, in either source. 

The Reviewer has erred in claiming that mothers’ IDs were matched between the HDSS and Household Survey. This was NOT the case. The only IDs that were matched were the Household IDs (HHID) at the onset of data collection and not at the time of analysis. Since the main aim of our study is to demonstrate that a cross-sectional sample household survey can be used to assess the quality of the data generated by a long-running and resource-intensive health and demographic surveillance system, matching IDs of mothers and children will be cumbersome as noted above in the case of wrongly entered DOBs. Again, the objective of the current study is such that matching children to their mothers is not a requirement. However, as we have noted in the Discussion section, correctly stated DOBs would have accorded us the opportunity to expand the analysis to include fertility by attempting to show that both data sources identified similar groups of women in terms of age-groups, and that they also reported similar child-bearing histories. 

-In fact, neither the survey nor the HDSS can be considered the most reliable source. However, the authors want to test whether the survey can be used to assess the data quality and reliability of the HDSS data (lines 278-279: p 12). To this question, given the limitations of the survey and the lack of matching of children, the answer is clearly no, even before any comparative analysis is done. Thus, I disagree with the conclusion (lines 344-345; p14) that "the results clearly demonstrate that an adapted DHS-type survey can be used to validate routine HDSS data". The authors mention another, more nuanced objective, which is to check whether the two sources "adequately match each other, with one hence validating the other" (lines 286-287; p12), but I doubt that they can even achieve this objective. I am fairly confident that some of the difference between the two sources is explained by operational definitions of residence and by missed and misplaced events due to the pitfalls of the two data collection procedures, but without a more rigorous comparison it is very difficult to determine which data collection bias leads to which bias in the mortality indicators.

We note Reviewer #2’s expressed opinions. We never attempted to claim that one data collection method is more reliable than the other. We are aware, and have clearly stated in the manuscript, that both methods have their advantages and disadvantages based on their operational definitions. We have also noted that in ideal circumstances where both methods collected accurate data, we would not expect the resulting datasets to be exactly the same. Our work was inspired by the fact that there is no established or recommended technique of assessing and confirming the validity of longitudinal data generated by HDSS platforms in LMICs, an our attempt represents a contribution towards methodological advances for resolving this major shortcoming for HDSS sites.

---

## [Decision Letter · Decision Letter 1]

1 Jul 2022

Assessment of the consistency of health and demographic surveillance and household survey data: A demonstration at two HDSS sites in The Gambia.

PONE-D-21-37201R1

Dear Dr. Jasseh,

We’re pleased to inform you that your manuscript has been judged scientifically suitable for publication and will be formally accepted for publication once it meets all outstanding technical requirements.

Kind regards,

Orvalho Augusto, MD, MPH

Academic Editor

PLOS ONE

Additional Editor Comments (optional):

Reviewers' comments:

Reviewer's Responses to Questions

**Comments to the Author**

1. If the authors have adequately addressed your comments raised in a previous round of review and you feel that this manuscript is now acceptable for publication, you may indicate that here to bypass the “Comments to the Author” section, enter your conflict of interest statement in the “Confidential to Editor” section, and submit your "Accept" recommendation.

Reviewer #2: All comments have been addressed

2. Is the manuscript technically sound, and do the data support the conclusions?

Reviewer #2: Partly

3. Has the statistical analysis been performed appropriately and rigorously? 

Reviewer #2: Yes

4. Have the authors made all data underlying the findings in their manuscript fully available?

Reviewer #2: Yes

5. Is the manuscript presented in an intelligible fashion and written in standard English?

Reviewer #2: Yes

6. Review Comments to the Author

Reviewer #2: Thank you for addressing most of my comments and suggestions. I now understand better the matching on HH ID only, given that the purpose was not to explain biases by comparing differences between the two sources.

However, I am still not comfortable with the conclusion: "This study attempts to ascertain whether a DHS-type cross-sectional household survey can be used to assess the quality of data and reliability of a prospective demographic surveillance system." To me, this means that a DHS-type survey can validate HDSS data, while the following sentence says something different: "...determining if the two different methods of data collection applied to the same population yield similar measures in terms of size, structure and commonly derivable childhood mortality indicators." The two sources could equally be right... or wrong (or anything in-between), while the biases of the two methods could balance out to yield the same results. Also, to say later that the two sources "adequately match each other" is not the same as "one validate the other".

7. PLOS authors have the option to publish the peer review history of their article (what does this mean?). If published, this will include your full peer review and any attached files.

Reviewer #2: No

---

## [Editor Report · Acceptance letter]

4 Jul 2022

PONE-D-21-37201R1 

Assessment of the consistency of health and demographic surveillance and household survey data: A demonstration at two HDSS sites in The Gambia 

Dear Dr. Jasseh:

I'm pleased to inform you that your manuscript has been deemed suitable for publication in PLOS ONE. Congratulations! Your manuscript is now with our production department. 

Kind regards, 

on behalf of

Dr. Orvalho Augusto 

Academic Editor

PLOS ONE